# Modular self-assembly of gamma-modified peptide nucleic acids in organic solvent mixtures

Sriram Kumar [1], Alexander Pearse [2], Ying Liu [1] & Rebecca E. Taylor [1,3,4 ✉]

Nucleic acid-based materials enable sub-nanometer precision in self-assembly for fields including biophysics, diagnostics, therapeutics, photonics, and nanofabrication. However, structural DNA nanotechnology has been limited to substantially hydrated media. Transfer to organic solvents commonly used in polymer and peptide synthesis results in the alteration of DNA helical structure or reduced thermal stabilities. Here we demonstrate that gamma-modified peptide nucleic acids (γPNA) can be used to enable formation of complex, self-assembling nanostructures in select polar aprotic organic solvent mixtures. However, unlike the diameter-monodisperse populations of nanofibers formed using analogous DNA approaches, γPNA structures appear to form bundles of nanofibers. A tight distribution of the nanofiber diameters could, however, be achieved in the presence of the surfactant SDS during self-assembly. We further demonstrate nanostructure morphology can be tuned by means of solvent solution and by strand substitution with DNA and unmodified PNA. This work thereby introduces a science of γPNA nanotechnology.

[1] Department of Mechanical Engineering, Carnegie Mellon University, Pittsburgh, PA, USA. [2] Department of Chemistry, Carnegie Mellon University, Pittsburgh, PA, USA. [3] Department of Biomedical Engineering, Carnegie Mellon University, Pittsburgh, PA, USA. [4] Department of Electrical and Computer Engineering, Carnegie Mellon University, Pittsburgh, PA, USA. ✉email: bex@andrew.cmu.edu

n the past 20 years, bottom–up manufacturing with DNA has emerged as a game-changing approach for creating the following structural paradigms: (1) DNA-based nanostructures like DNA origami[1–3], (2) programmable materials whose multiscale assembly is directed by DNA binding[4,5], and (3) hybrid top–down/bottom–up systems that leverage traditional lithographic microfabrication alongside self-assembly processes[6–8]. These systems benefit from the robust sequence complimentarity and specificity of DNA. However, DNA nanostructures are dependent on high salt concentrations for structural stabilization, susceptible to enzymatic degradation, and undergoes denaturation in organic solvents[9–12]. This limits the capability of unprotected DNA nanostructures for realizing many applications that require robust structural stability and transferability to other systems. For example, processes used in polymer synthesis[13] and peptide synthesis[14,15] often utilize polar aprotic solvents like dimethyl formamide (DMF) and dimethyl sulfoxide (DMSO) that can cause denaturation, reduced thermal stability, and conformational changes in DNA duplexes[12,16,17]. To create protected DNA nanostructures that can operate in such solutions, strategies including base-specific cross-linking[18], coating with protective molecules[19], and encapsulation within protective structures[20,21] can be employed.

However, nanostructures capable of forming within DMF and DMSO solutions could introduce the reliable sub-nanometer structural control of nucleic acid nanotechnology into broad fields like polymer synthesis, in which sequence-dependent structural control remains a challenge[10,22]. Previous studies aiming to mimic the self-assembling properties of DNA have investigated nucleobase-containing polymers, DNA–synthetic polymer conjugates, and synthetic DNA mimics with altered backbones called xeno nucleic acids[10,23,24]. As described by Wilks et al.[23], conjugation of DNA to synthetic polymers through a wide range of chemistries often results in less than desirable yield of conjugate products in organic solvents, with decreasing yield upon introducing more hydrophobic polymers. Strategies to synthesize nucleobase-containing polymers are currently limited by the lack of specific sequence control and the need for novel nucleobase-containing molecular structures, including monomers, oligomers, or polymers[10,22]. Synthetic nucleic acid mimics, or xeno nucleic acid strategies, currently hold the most promise for translating nucleic acid nanotechnology into organic solvents[17,25], with peptide nucleic acid (PNA) as a popular candidate among them.

PNA was first presented in the early 1990s as a novel strategy to generate artificially synthesized biopolymers by Nielsen et al. and Egholm et al.[26,27]. Their backbones consist of uncharged repeats of N-(2-aminoethyl) glycine units (aeg) linked by peptide bonds. PNAs exhibit many interesting properties, including high binding affinity to DNA and RNA, a low dependency on ionic strength, high chemical stability, high sequence specificity, and resistance to both nucleases and proteases[28].

Unlike the sugar-phosphate backbone of DNA, the backbone of PNA is not inherently negatively charged, which gives PNA a peptide-like tendency to aggregate and non-specifically adhere to surfaces and macromolecules[29,30]. For this reason, no complex PNA structures defined by Watson–Crick base pairing have been reported. Studies toward the generation of nanoscale and micronscale PNA-based assemblies have instead typically been based on conjugation to lipophilic polymers[31], hybridization of PNA segments to DNA nanostructures using modified annealing protocols to prevent aggregation[32], and reprogramming self-assembly via hybridization to specific small molecules like cyanuric acid[33]. A more recent advance in the efficient and rapid formation of nanostructures from unmodified PNA was developed by Berger et al. They demonstrated that simple guanine-containing PNA

monomers and PNA sequences with two side chains can form ordered structures under alkaline conditions[24,34]. However, a modular approach to self-assemble multiple PNA oligomers via programmed complementarity remains largely unanswered.

PNA is a versatile material, and recent modifications have enhanced its potential as a modular building material. Specifically, efforts to reduce PNA aggregation[35–38] and make this molecule cell permeable[27,39,40] have resulted in numerous new PNAs with altered solution shape and solubility[41–44]. In 2006, Dragulescu-Andrrasi et al. reported that a simple modification within the gamma position of N-(2-aminoethyl) glycine backbone of PNA caused the single-stranded molecule to assume a preorganized helical arrangement[45]. This molecule is called γPNA, and it can bind to DNA and RNA with exceptionally high affinity and sequence selectivity[46]. While the development of γPNA was largely aimed at improving anti-sense therapies and molecular diagnostics[47,48], we hypothesized that the resulting higher binding affinity due to the pre-organized helical arrangement of (R)-diethylene glycol (mini-PEG) containing γPNA would enable it to be used for the formation of complex nucleic acid nanostructures in organic solvent mixtures. Here we show the design considerations and self-assembly of a γPNA 3-helix nanofiber in select organic solvent mixtures using distinct γPNA oligomers. This work also investigates the role of different solvent mixtures, oligomer substitution with DNA, and unmodified PNA, as well as the use of anionic surfactant sodium dodecyl sulfate (SDS) to regulate morphology of nanostructures.

## Results

**Concept and design of γPNA 3-helix fiber.** We present here a structural motif for building periodic nanofibers with nine unique γPNA strands. Our design is adapted from the single-stranded tile (SST) approach in DNA nanotechnology[49–51]. Forming DNA SST nanofibers generally requires only a few distinct oligonucleotide species (4–20) that can polymerize and grow to become multiple microns in length[50]. In addition, we also prescribe to the nomenclature of nanofibers for our SST-based nanostructure within this article wherein the field of DNA nanotechnology have used terms, such as helix bundle[51].

Figure 1a shows the key structural difference between PNA and DNA double helices. B-form DNA are right-handed double helices that rotate 34.3° per base pair or 10.5 base pairs per helical turn[52,53]. To account for this property and prevent undesired prestress, DNA SST designs usually aim for 10.4–10.7 bases per helical revolution[54,55]. Unlike DNA, PNA double helices are reported to have 18 base pairs per turn[56]. Therefore, we chose to design an SST motif with 18 bases long repeat unit to make a nanofiber consisting of three interwoven double helices. The design reported here (Fig. 1b) is based on repeating tubular units where the number of unique oligomers required to form the individual units are 3× the number of helices in the corresponding bundles (i.e., 9 oligomers for 3-helix nanofiber). The constituent strands fall into one of the two categories: two-thirds of them (6 oligomers, Fig. 1b magenta and cyan arrows) are contiguous strands that are arranged linearly and the other one-third (3 oligomers, Fig. 1b green arrows) are crossover strands that each form a crossover from one helix to a neighboring helix. Each γPNA sequence contains 3 gamma modifications with (R)-diethylene glycol (mini-PEG) at the 1, 4, and 8 positions (Fig. 1b, gray dots). In order to immobilize and visualize γPNA nanostructures using fluorescence microscopy, N-terminal functionalization of select strands with biotin (Fig. 1b, orange oval) and Cy3 (Fig. 1b, pink star) was performed. Each oligomer consist of 12 bases, which follows a 6 + 6 domain-binding pattern. For PNA double helices with 18 base pairs per full helical turn, 6 base

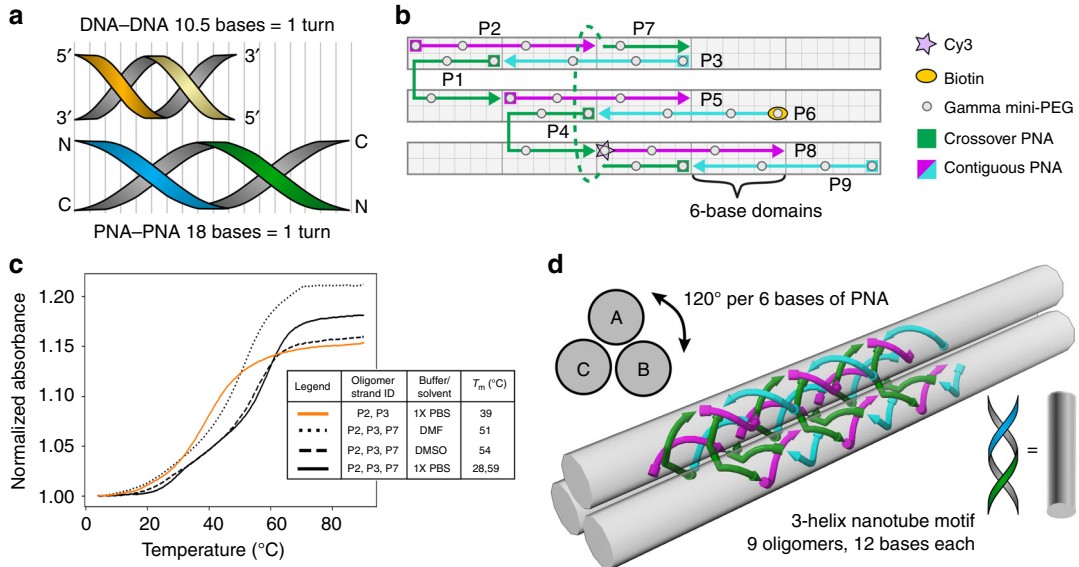

**Fig. 1 The design of 3-helical γPNA nanofibers. a** Schematic comparison between a full helical turn of DNA and PNA double helix. B-form DNA is typically stable with 10.5 bases per helical turn while PNA has a helical pitch of 18 bases per turn. **b** A schematic representation of the structural motif for the SST design that shows 6-base domains and an overall repeat unit of 18 bases. Each PNA oligomer is modified at the 1, 4, and 8 γ-positions with (R)-diethylene glycol (gray dots) to enable pre-organized helical conformation. γPNA oligomers are classified into contiguous (magenta and cyan arrows) or crossover (green arrows) PNAs based on their position in the motif. Specific γPNA oligomers (P8 and P6) are labeled with fluorescent Cy3 (pink star) and biotin (orange oval), respectively, to enable detection of structure formation using fluorescence microscopy. **c** Melt curve studies show the melting temperatures of 2-oligomer (orange) and 3-oligomer (black) substructures in different solvent conditions vary between 39 and 59 °C. This indicates that 6-base domains are sufficiently stable for γPNA–γPNA systems in PBS (solid curve), DMSO (dashed curves), and DMF (dotted curves). Source data are provided as a Source data file. **d** 6-Base domains correspond to 120° rise in helical rotation enabling the structural motif to program the assembly of 3-helix nanofibers that can polymerize lengthwise.

pairs correspond to 120° rise in helical rotation in a triangular-sectioned nanofiber (Fig. 1b, d).

For thermal stability, most DNA-based SST systems use domains of 10 or 11 bases[50,51]. To verify that 6-base γPNA domains would bind sufficiently strongly, we measured the melting temperatures with 2- and 3-oligomer γPNA–γPNA systems using 6-base domains (Fig. 1c—orange and black solid lines) and compared them to studies of DNA–DNA systems in aqueous buffers from literature (Supplementary Fig. 1). These studies demonstrate that the thermal stability of 6-base γPNA domains are similar to or exceed that of 10-base DNA domains (See Supplementary Fig. 1). In addition, the 3-oligomer γPNA–γPNA system in the aqueous buffer condition (Fig. 1c—black solid line) showed two transitions in its melt curve. The lower temperature transitions evident in the three-stranded systems are attributed to melting of helical structure within the overhanging γPNA domains. To substantiate this prediction, we measured melting temperatures of the associated single-stranded γPNAs with the same buffer conditions, in which two of the associated oligomers showed melting temperatures around 28 °C (Supplementary Fig. 2).

Melt curve studies for these 3-oligomer systems were also performed in organic solvents including DMSO and DMF as shown in Fig. 1c. These results indicate γPNA duplexes formed successfully in organic solvents and experienced only minor reductions in melting temperatures in DMSO and DMF as compared to aqueous buffer formation (Fig. 1c—dotted and dashed black lines). This is in stark contrast with assemblies of short DNA oligomers, which are denatured in DMSO[11]. This also concurs well with the results previously published by Sen et al.[17] on the effects of organic solvent mixtures during lysine-tagged aeg-PNA annealing. Unlike their DNA counterparts, organic solvent mixtures have a much smaller effect on the thermal

stability of PNA–PNA duplex because the destabilization of DNA duplexes in an aprotic solvent is assumed to be predominantly caused by dehydration and ion exclusion[57].

Comparing melt curve studies between unmodified aeg-PNA and γPNA duplexes for isosequential 2- and 3-oligomer systems in aqueous buffer as well as DMF and DMSO verified that both aeg-PNA and γPNA systems experience either minor or no reductions in melting temperatures (Fig. 1c and Supplementary Fig. 3a). However, the isosequential 3-oligomer γPNA system shows considerably higher melting temperatures in comparison to the 3-oligomer aeg-PNA system in aqueous buffer as well as DMF and DMSO solvents. This is consistent with the stronger binding affinity and higher thermal stability resulting from the conformation-enhancing γ-modifications in γPNA systems not present in aeg-PNA systems. This also agrees with previously published work by Sahu et al. that compared PNA–DNA and PNA–RNA duplex thermal stability between the unmodified aeg backbone and the mini-PEG γ-modified backbone[46]. They showed that the incorporation of a single mini-PEG side chain stabilized a PNA–DNA duplex by 2.3–4 °C. Further, Sobczak et al. reported that DNA nanostructures with multiple DNA oligomers fold cooperatively at temperature ranges higher and narrower than the melting temperatures of associated individual domains[58]. Given that we find an increase in melting temperature from 7 to 19 °C for our 2- to 3-stranded γPNA systems (Fig. 1c), we expect that 3-helix SST nanofibers will have further increased thermal stability than 2- and 3-oligomer systems (see Supplementary Table 3).

In total, 9 oligomers make up this γPNA nanofiber structural motif, and each oligomer is 12 bases long. The structural motif programs the self-assembly of 3-helix nanostructures that can polymerize along lengthwise (Fig. 1d). This structure is notable in being made of SSTs with only two domains each. A resulting

property of this design is that theoretically all sequences must be present and successfully bound in order to enable polymerization. In other words, structural formation indicates that every oligomer has been successfully incorporated.

**Evidence of γPNA self-assembly.** Favorable conditions for self-assembly of γPNA oligomers were thereafter determined by screening combinations of various temperature anneal ramp cycles [0.5–0.1 °C min$^{-1}$ and 0.1 °C for 1–3 min$^{-1}$], various functional additives [5–30% (wt V$^{-1}$) PEG8000, 10–40% (V V$^{-1}$) formamide, 1–8 M urea in varying concentrations], and a variety of solvent mixtures including organic solvents like DMF [10–87.5% (V V$^{-1}$)], DMSO [10–87.5% (V V$^{-1}$)], acetonitrile [ACN; 10–50% (V V$^{-1}$)], and primary alcohols [10–50% (V V$^{-1}$) methanol, ethanol]. These systems were then characterized using total internal reflection fluorescence (TIRF) microscopy to visualize structure formation. While self-assembly was observed in organic solvent mixtures such as 75% DMF:H$_2$O (V V$^{-1}$ %) and 40% 1,4-dioxane:H$_2$O (V V$^{-1}$ %), the microscopic observation of well-organized architectures was most evident from our TIRF studies when oligomers were annealed with a slow temperature ramp (Supplementary Table 4) in 75% DMSO:H$_2$O (V V$^{-1}$ %) as shown in Fig. 2a (Supplementary Fig. 5).

In contrast, no structures were visible under TIRF assays when the system of oligomers introduced had one missing γPNA sequence or a mismatched γPNA sequence indicating the Watson–Crick base pair-driven self-assembly of these micron-scale structures. Sahu et al. had previously shown the high sequence selectivity of mini-PEG γPNA to be less accommodating to structural mismatches than even unmodified aeg-PNA[46]. Moreover, replacing the entire system with sequence-identical DNA oligomers also did not show structure formation in 75% DMSO:H$_2$O, 1× phosphate-buffered saline (PBS; a physiological buffer) or 1× TAE plus 12.5 mM MgCl$_2$ (a typical DNA SST buffer) for similar and slower annealing ramps largely indicating that this structure formation is due to the large cooperativity, specificity, and stability of conformationally pre-organized γPNA oligomers. For further substantiation, we sequentially introduced increasing amounts of aeg-PNA up to 7 oligomers. Systems containing combinations of up to 4 aeg-PNA oligomers under similar conditions of 75% DMSO:H$_2$O continued to form nanofibers robustly. However, when several combinations of oligomer sets containing 5–7 aeg-PNA oligomers were included, we observed formation of aggregates and no nanostructure formation using TIRF assays (Supplementary Fig. 3b).

In addition, a histogram of contour length profile measurements (see Supplementary Note 1) of self-assembled γPNA in

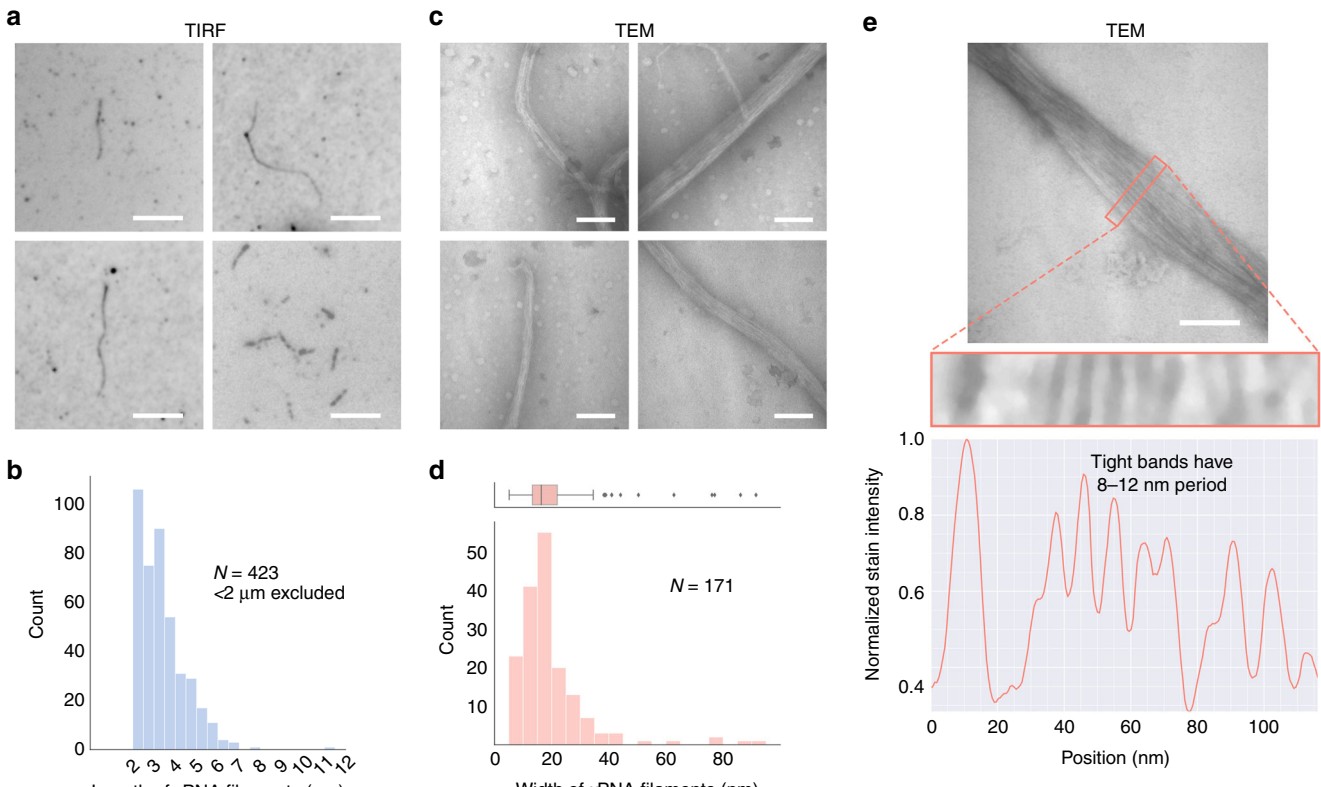

**Fig. 2 Evidence and characterization of self-assembly of γPNAs to form 3-helix nanofibers using TIRF and TEM assays. a** TIRF panels (5 μm scale bar) of γPNA self-assembly in 75% DMSO:H$_2$O (V V$^{-1}$) obtained through monitoring the Cy3 channel provides evidence of well-organized, micron-scale filamentous structures. **b** The length distribution of 3-helix tube structures (sample size, N = 423 visibly separate nanostructures over 3 independent experiments) self-assembled in 75% DMSO:H$_2$O is thresholded to include elongated nanostructures at least 2 μm in length. While short filaments are more prevalent, filaments up to 11 μm in length were also observed. Source data are provided as a Source data file. **c** TEM panels (100 nm scale bar) of γPNA nanostructures annealed in 75% DMSO:H$_2$O (V V$^{-1}$) shows bundling of γPNA nanostructures at the nanoscale level along its width. **d** Width distribution of γPNA nanostructures (sample size, N = 171 visibly separate nanostructures over 4 independent experiments) using TEM studies shows a right skew with a median width of 16.4 nm, IQR of 7.2 nm, and maximum values beyond 80 nm. This suggests that nanostructures bundle together when annealed in 75% DMSO:H$_2$O (V V$^{-1}$). Source data are provided as a Source data file. **e** Line profile scan along the width of any bundled γPNA nanofibers (repeated over four independent experiments) shows the presence of thinner γPNA substructures occurring between periods of 8 to 12 nm, which agrees with constituent structures of 5–6 nm diameter as predicted by our motif.

75% DMSO:H$_2$O shows that nanostructures form multi-micron filaments, with some structures reaching ~11 µm (Fig. 2b). These lengths compare well with existing DNA SST strategies to construct nanofibers[49–51]. Transmission electron microscopic (TEM) imaging of the $\gamma$PNA system in the 75% DMSO:H$_2$O solvent mixture confirmed the formation of nanofibers at nanoscopic resolutions providing proof of concept but showed further features that suggest bundling of nanofibers (Fig. 2c).

Width profiles of TEM images (see Supplementary Note 2) shows a right skew with a median width of 16.4 nm and maximum values beyond 80 nm providing further evidence that the $\gamma$PNA nanofibers in 75% DMSO:H$_2$O has tendencies to bundle along their widths (Fig. 2d). Based on previously published $\gamma$PNA–DNA[59] and PNA–DNA[60] helix diameters of 2.3 nm, we estimated that individual 3-helix nanofibers would have diameters of 5–6 nm. Our experimental findings thus indicate that, although $\gamma$PNA nanofibers can form, existent hydrophobic effects around these structures still need accounting for through a more exploratory search of solvents, pH, and chemical and charge modifications to enable monodisperse nanofiber populations. Line profile analysis (see Supplementary Note 2) across the widths of these "bundled" nanofibers, however, show alternating bands of light and dark, with tighter alternating bands occurring in 8–12 nm periods, consistent with substructures that have diameters in the 5–6-nanometer range (Fig. 2e).

Furthermore, at the microscale $\gamma$PNA nanostructures show a different, more spicular shape when annealed using the same thermal anneal ramp in 75% DMF:H$_2$O and sparse structure formation in 40% 1,4-dioxane:H$_2$O (V V$^{-1}$) indicating the role of solvent in modifying the bundling characteristics or even potentially the Watson–Crick base pair interactions between multiple $\gamma$PNA oligomers (Supplementary Fig. 4). This finding is consistent with previous work showing that organic solvents such as DMF can suppress hydrogen bonding interactions in self-assembling nucleobase-containing polymers, thereby affecting their shape and morphology[61].

**Capability to form $\gamma$PNA-DNA hybrid structures in organic solvent mixtures**. To investigate the capacity to form hybrid $\gamma$PNA-DNA structures, analogous DNA oligomers were selectively introduced into $\gamma$PNA nanostructures. DNA oligomers can introduce novel functionalization, alter charge and potentially increase hydrophilicity. For example, replacing P3 with a fluorescein-labeled DNA enabled colocalization studies shown in Supplementary Fig. 6.

In order to explore the role of DNA in modifying hydrogen-bonding and hydrophobic effects influenced by organic solvent mixtures, we introduced through sequential replacement several combinations of up to three $\gamma$PNA oligomers with their equivalent unmodified DNA oligomers. Such an investigation could lead to a better understanding of contributions from hydrophobic forces that affect nanofiber structure formation and the thermodynamic feasibility of $\gamma$PNA–DNA hybrid nanostructures due to the introduction of a less stable $\gamma$PNA–DNA duplexes during self-assembly. As shown in Fig. 3, we sequentially replaced both the contiguous (P3, P5, P9) and crossover (P1, P4, P7) $\gamma$PNA oligomers with their corresponding DNA oligomers. In all cases, the introduction of DNA did not prevent nanostructure formation, and structures were visible under TIRF microscopy. Contour length profile measurements made from TIRF studies of the contiguous $\gamma$PNA replacements with DNA showed proportionally sparser fields of filaments for constructs >2 µm (Supplementary Fig. 7).

Interestingly, the replacement of contiguous $\gamma$PNA oligomers with DNA resulted in straight filaments (Fig. 3a–c, Supplementary

Fig. 8a–c), suggesting that $\gamma$PNA–DNA nanostructures adopt different morphologies upon the introduction of oligomers that carry a different charge and helical conformation. In parallel, the replacement of the same contiguous $\gamma$PNA oligomers with the corresponding aeg-PNA oligomers showed no discernible difference in comparison to all-$\gamma$PNA nanofibers under TIRF assays (Supplementary Fig. 3c). While the mechanisms driving the differences in morphology between contiguous DNA and aeg-PNA replacements are unclear, differences may be related to the pre-organization of $\gamma$PNA backbone. If helical properties of $\gamma$PNA and DNA determine the nanostructure morphologies, then previous studies of $\gamma$PNA–DNA duplexes can be used to predict twist in a hybrid structure. Specifically, given that $\gamma$PNA–DNA helical pitch has been reported to 15 bases per turn as opposed to 18 base pairs per turn found in $\gamma$PNA–$\gamma$PNA nanostructures[59]. Incorporation of DNA into a structure would cause a global right-handed twist. However, given the lack of evidence for twist (Fig. 3a–c), in the context of a complex nanostructure, complimentary $\gamma$PNA strands may not have the flexibility or the conformational freedom to accommodate corresponding DNA oligomers. Under such circumstances, DNA oligomers would have to undergo a conformational change of their own for hybridization to take place. This has been previously theorized by Sahu et al. in the context of $\gamma$PNA strands with a high density of $\gamma$-modifications with respect to $\gamma$PNA–DNA duplex formation[46].

In contrast, the introduction of crossover DNA oligomers promoted more pronounced bundling of these nanofibers, with stellate structures visible under TIRF and TEM (Fig. 3d–f). It is particularly interesting to note that the geometrical position of oligomeric replacement with DNA in the SST motif has a direct effect on the shape and morphology of the resultant structure. Moreover, the stellate structure morphology continues to persist albeit to a proportionally lesser degree, when the same crossover $\gamma$PNA oligomers are substituted with aeg-PNA oligomers (see Supplementary Fig. 3d). This pronounced bundling in both DNA and aeg-PNA crossover substitution case may be the result of a combination of effects. It may depend on (1) the degree of surface and solvent exposure of the replacement oligomers affecting the overall hydrophobicity of the self-assembled architecture and (2) helical form changes associated with shorter per helix DNA–$\gamma$PNA-binding regions of crossover oligomers. Our TIRF studies suggest that, while the structure formation is primarily driven by Watson–Crick hydrogen bonding, secondary forces such as overall hydrophobic effects may affect structural morphology adopted by the nanofibers.

Furthermore, width distribution for contiguous DNA replacement constructs show median widths similar to all-$\gamma$PNA constructs (Supplementary Fig. 9). This indicates that localized improvements to surface hydrophilicity through charge modifications provides little stabilization to bundling tendencies for these constructs, further indicating a need for more uniform surface modifications. TEM images of the crossover-substituted $\gamma$PNA–DNA tubes also suggest looser weaving or perhaps edge fraying that may be related to a partially disrupted crossover architecture (Fig. 3d–f, TEM panels).

**Influence of SDS on bundling of $\gamma$PNA nanofibers during self-assembly**. We investigated the regulatory effects of anionic surfactant SDS on the morphology of $\gamma$PNA nanofibers during self-assembly in 75% DMSO:H$_2$O (V V$^{-1}$), specifically its effects toward reduced bundling. Several studies have focused on the modulation of peptide and protein conformation through surfactant addition. In particular, the effects of SDS have been shown to stabilize fibril formation in amyloid $\beta$-type structures below its critical micelle concentration (CMC; 8.2 mM)[62,63]. The addition

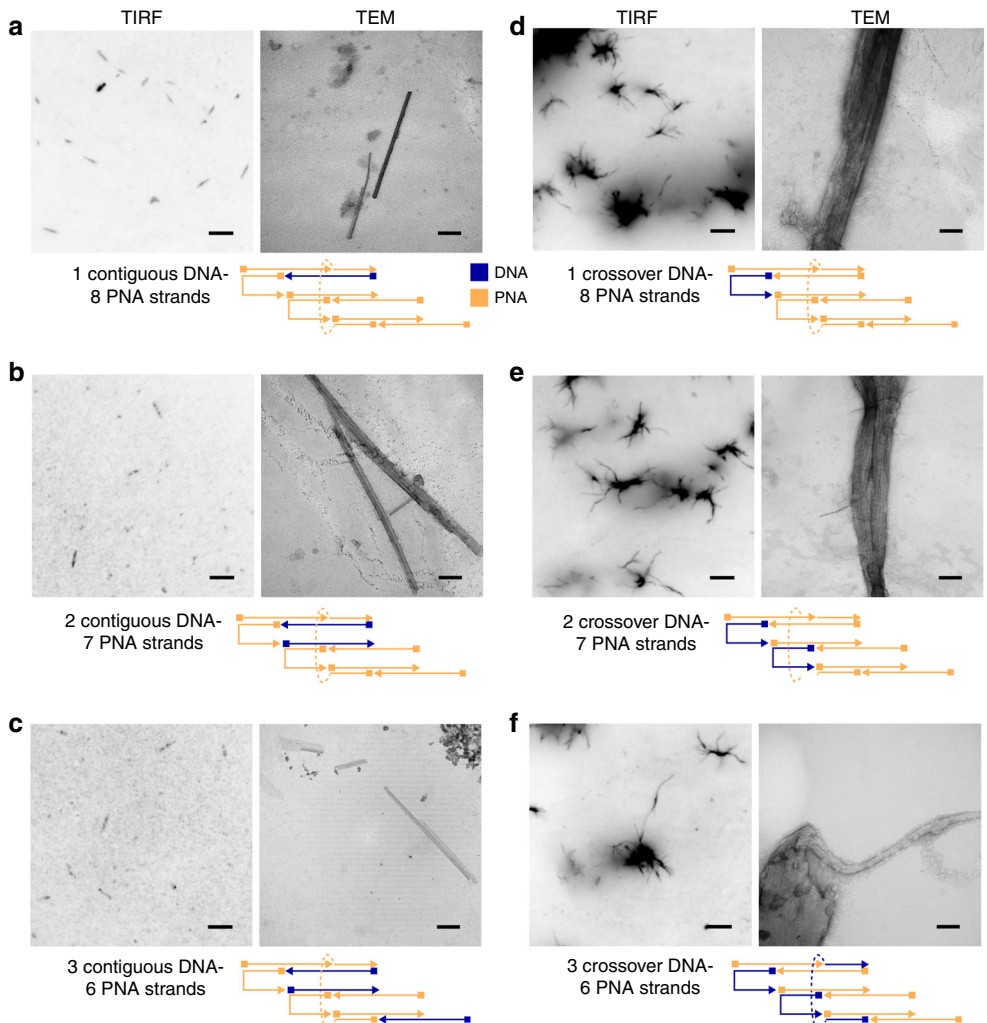

**Fig. 3 Self-assembly and characterization of γPNA–DNA hybrid nanofibers.** TIRF and TEM characterization of γPNA–DNA hybrid filaments through selective and sequential replacement of γPNA oligomers with DNA. Schematic representations (repeated over two independent experiments for each condition shown in **a**–**f**) show the position in the SST motif replaced with DNA (blue arrows) in the context of other γPNA sequences (orange arrows). Sequential replacement of: **a** one contiguous γPNA sequence with DNA, **b** two contiguous γPNA sequences with DNA, and **c** three contiguous γPNA sequences with DNA resulted in straight filaments. However, sequential replacement of: **d** one crossover γPNA sequence with DNA, **e** two crossover γPNA sequences with DNA, and **f** three crossover γPNA sequences with DNA resulted in stellate structures with pronounced bundling effects. Scale bars on TIRF images are 5 μm, and scale bars on TEM images are 100 nm.

of SDS was also found to initiate morphological transitions in amphiphilic peptides[64]. These transitions induced by anionic surfactants have been mainly attributed to the effect of hydrophobic and electrostatic interactions between surfactants and peptide molecules. In addition, previous literature has shown that SDS can promote hydrogen bonding in otherwise unstructured short peptides and allows adoption of secondary structures such as β-sheets[65]. We therefore hypothesized that the use of SDS would not disrupt the hydrogen bonding in Watson–Crick base pairing of γPNA oligomers and could promote reduced bundling over a specified range of concentrations.

We therefore studied the effects of increasing SDS concentration both below and above its CMC toward reduced γPNA nanofiber bundling using TIRF assays. As seen in Fig. 4a, TIRF panels show that, upon increasing SDS concentration up to 5.25 mM, thinner morphologies of nanofibers based on fluorescence intensity become more dominant. This may indicate that the SDS-induced development of a net charge across our nanofiber structure with increasing SDS concentrations results in reduction of bundling of γPNA nanofibers. To verify this

capability of SDS to reduce non-specific interactions between nanofibers, we performed TIRF assays on γPNA crossover replacements with both their equivalent DNA and aeg-PNA crossover oligomers in the presence of 5.25 mM SDS. As highlighted before, we previously observed stellate-like morphologies in both cases in the absence of SDS. However, in the presence of SDS, we were able to visually observe the disappearance of stellate-like morphologies, suggesting that electrostatic interactions introduced by SDS in the system are able to counter the increased non-specific interactions caused by overall hydrophobic effects in hybrid nanofibers (see Supplementary Fig. 10). In addition, when concentrations of SDS neared or exceeded the CMC concentration, appearance of largely networked γPNA nanofibers occur with increasing propensity (Fig. 4a). This is consistent with previous work by Cao et al. showing that SDS induces higher-order peptide assemblies when present in high concentrations[64].

TEM imaging of the all-γPNA co-self-assembly in the 75% DMSO:H$_2$O solvent mixture along with 5.25 mM SDS confirmed the formation of nanofibers with diameters predominantly

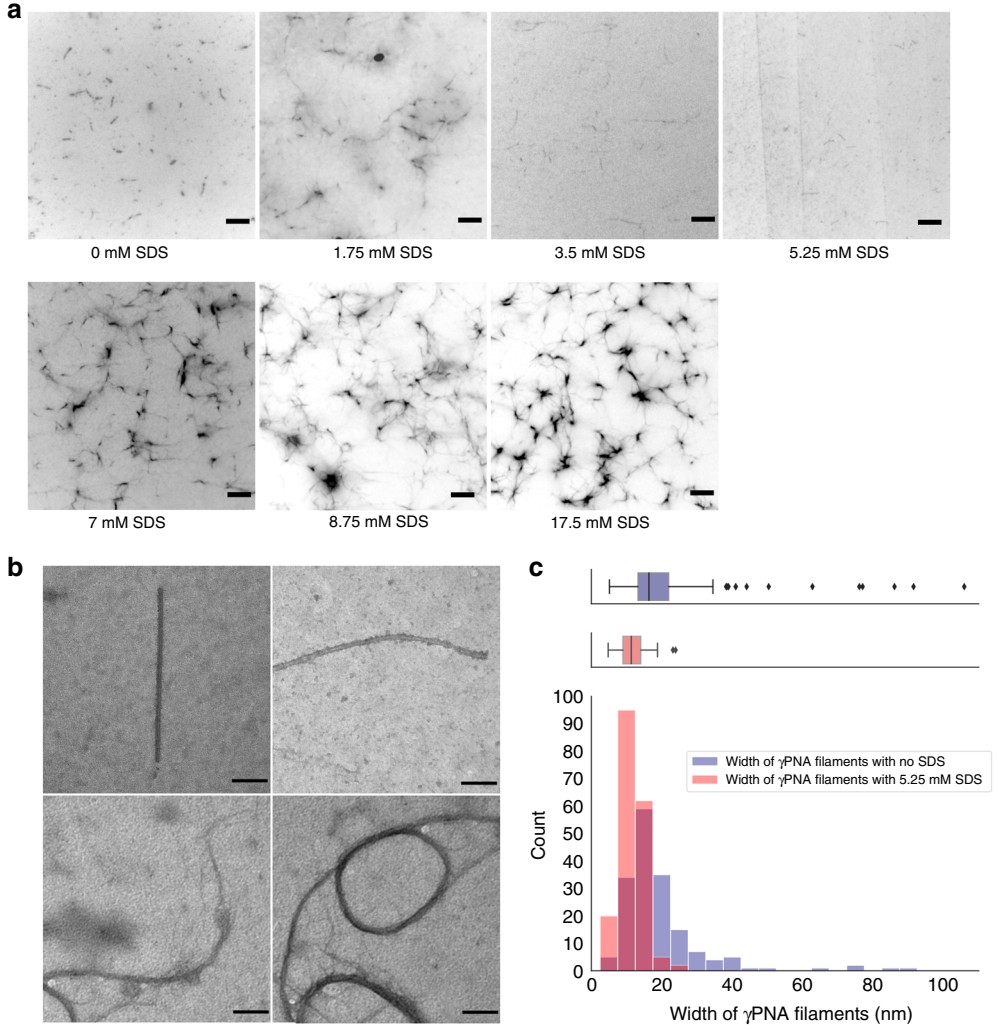

**Fig. 4 Effect of SDS on the width of γPNA nanofibers during self-assembly using TIRF and TEM assays. a** TIRF panels (5 μm scale bar) of the self-assemblies formed by γPNA in 75% DMSO:H$_2$O (V V$^{-1}$) with different concentration of SDS ranging from 0 to 17.5 mM (repeated over 2 independent experiments for each condition). With increasing concentrations of SDS [0–5.25 mM], thinner morphologies of the nanofibers become more dominant. When concentrations of SDS neared or exceeded the CMC concentration (8.2 mM), networked morphologies of γPNA with increasing propensity was observed. **b** TEM panels (100 nm scale bar) of γPNA nanostructures annealed in 75% DMSO:H$_2$O (V V$^{-1}$) with 5.25 mM SDS shows nanofibers with diameters of 8–12 nm range. **c** Overlay of width distribution of γPNA nanostructures with 5.25 mM SDS (sample size, $N = 185$ visibly separate nanostructures over 2 independent experiments, red) and without SDS (sample size, $N = 171$ visibly separate nanostructures over 4 independent experiments, blue) using TEM studies. Nanofibers in the presence of 5.25 mM SDS show a tight distribution with a median width of 11.3 nm and IQR of 5 nm. Source data are provided as a Source data file.

between 8 and 12 nm at nanoscopic resolutions (Fig. 4b). Width profile measurements made using TEM imaging confirmed that γPNA nanofibers in the presence of 5.25 mM SDS have a tight width distribution with a median width of 11.3 nm (Fig. 4c). Therefore, the addition of anionic surfactants like SDS provides a simple and effective way for regulating the self-assembly morphology of γPNA nanofibers by changing the surfactant concentration.

## Discussion

We report here the modular self-assembly of the nanomaterial γPNA to form nanofiber structures using the nucleic acid SST strategy. This work demonstrates that conformation-enhancing modifications of γPNA provide high cooperativity, thermal stability, and specificity that enable robust nanostructure formation. While in several cases the introduction of >50% aeg-PNA to the oligomer set led to the formation of aggregates, this result may be

specific to the SST structural motif, size, and solvent conditions associated in this study. Suitable design of the structural motif accounting for the reduced cooperativity and thermal stability of aeg-PNA in comparison to γPNA and tuning thermal ramp rates, solvent choice, pH, and charge modifications to counter kinetic traps, hydrophobic effects and solubility might enable robust structure formation at the nanoscale.

The solvophilic capability of γPNA to form nanostructures in organic solvent mixtures demonstrates that γPNA nanostructures can extend the range and utility of nucleic acid nanotechnology to environments beyond substantially hydrated media. Our studies show that, in the presence of appropriate concentrations of anionic surfactants like SDS, γPNA oligomers can form nanofiber populations with a tight width distribution.

This study has focused on the design and formation of one-dimensional, micron-scale filaments using a specific SST structural motif and specific type and density of mini-PEG

$\gamma$-modifications on PNA. Therefore, future studies are needed to investigate additional structural motifs and the effects of type and density of gamma functionalization on these systems to expand toward a three-dimensional architectural space. One example of this includes recent work from the Heemstra group on the self-assembly of micellar architecture using $\gamma$PNA with specific amino acid $\gamma$-modifications showing the capability to encode bilingual behavior through protein and nucleobase codes toward more complex nanoarchitectures[66]. In this work, we have also demonstrated that, while the formation of hybrid $\gamma$PNA–DNA nanostructures would be primarily driven by Watson–Crick base pairing, the structural morphology may be the result of a combination of effects. Specifically, the resultant morphology is affected by the degree of surface and solvent exposure of the DNA oligomers affecting the overall hydrophobicity of the SST motif and the helical form changes associated with the shorter per helix DNA–$\gamma$PNA-binding regions of contiguous and crossover regions. In future studies, it is therefore important to investigate the pitch- and stability-related differences arising due to the structural differences between $\gamma$PNA–$\gamma$PNA, $\gamma$PNA–DNA, and DNA–DNA. These investigations will require the development of new $\gamma$PNA-specific structural motifs for building a broader range of functional structures.

This work additionally provides a proof-of-concept demonstration that micron-scale all-$\gamma$PNA and hybrid $\gamma$PNA–DNA nanostructures can be formed in organic solvent mixtures that meet the following criteria: (1) solvent mixtures are sufficiently polar to retain solubility of the $\gamma$PNA–DNA complexes even above 50% organic solvent[17]; (2) solvent mixtures are aprotic or otherwise have reduced hydrogen-bonding donor/acceptor activity of the solvent to promote nucleobase hydrogen bonding[57]; and (3) solvent mixtures have a high boiling point to allow for slow thermal ramp annealing in the context of multiple unique oligomers to avoid kinetic traps or misfolding. To understand the potential of this material, future studies are needed to determine whether $\gamma$PNA can replicate the complexity and capability of DNA as a nanomaterial.

## Methods

**Materials**. Unmodified and high-pressure liquid chromatography (HPLC)-purified modified DNA were purchased from IDT DNA. All diethylene-glycol-containing $\gamma$-modified PNA were obtained from Trucode Gene Repair, Inc. (Woburn, MA). Oligomer sequences are shown in Supplementary Table 1. Polar aprotic solvents like DMF, DMSO, and ACN were purchased in their anhydrous form from Sigma-Aldrich. All aqueous buffers were prepared in-house with chemicals like NaCl, KCl, SDS, $Na_2HPO_4$, and $KH_2PO_4$ purchased from VWR.

**Solid-phase PNA synthesis**. The aeg-PNA monomers were purchased from Polyorg Inc. (Leominster, MA) and used without further purification. PNA sequences (see Supplementary Table 2) were synthesized using the solid-phase tert-butyloxycarbonyl (Boc)-protection peptide synthesis strategy[67]. Oligomers were synthesized off of p-methyl-benzhydrylamine resin·HCl (0.45 mequiv g$^{-1}$, Peptides International). The number of active amine sites on the resin was lowered to 0.1 mequiv g$^{-1}$ by coupling the first monomer of the respective sequence to the resin at 0.1 mmol and then capping unreacted amine sites by acetic anhydride (Sigma-Aldrich). Boc-PNA were coupled to the resin using 2-(1$H$-benzotriazol-1-yl)-1,1,3,3-tetramethyluronium hexafluorophosphate (379 mg, 1.0 mmol, HBTU, Chem-Impex) with $N,N$-dicyclohexylmethylamine (Sigma-Aldrich) as the base, respectively. Qualitative Kaiser tests were performed to assess resin deprotection and successful monomer coupling. Oligomers were cleaved from the solid support using m-cresol/thianisole/trifluoromethanesulfonic acid/trifluoroacetic acid (TFA) (1:1:2:6). PNA is precipitated using cold diethyl ether. Purification of the PNA oligomers was performed by reverse-phase HPLC with a C18 silica column on a Waters 600 controller and pump with a Waters 2996 photodiode array detector to monitor absorbance changes (Supplementary Fig. 11).

Characterization of aeg-PNA oligomers was performed using matrix-assisted laser desorption/ionization coupled to time-of-flight (MALDI-ToF) mass spectrometry (MS) on an Applied Biosystems Voyager biospectrometry workstation using $\alpha$-cyano-4-hydroxycinnamic acid as the matrix (10 mg mL$^{-1}$ in water/ACN, 0.1% TFA). MALDI-ToF spectra are shown in Supplementary Fig. 12.

LC-MS data of $\gamma$PNA oligomers have been shown (with permission from Trucode Gene Repair, Inc.) in Supplementary Figs. 13–33.

**Melting experiment assay**. Variable temperature ultraviolet–visible experiments were performed in a Varian Cary 300 spectrophotometer equipped with a programmable temperature block in 1 cm optical path, quartz cells. The melting curves were primarily recorded using the Varian Cary thermal software over a temperature range 15–90 °C for both cooling (annealing) and heating (melting) cycles at a rate of 0.5 °C min$^{-1}$. The lower temperature was sometimes extended to 4 °C. The samples were kept for 10 min at 90 °C before cooling and at 4 or 15 °C before heating. The melting temperature ($T_m$) was determined from the peak of the first derivative of the heating curve.

Thermal melting curves in 100% DMF show severe noise or signal disturbances partly because of high absorbance of DMF at the wavelength range used during experimentation. This is a noted phenomenon in literature[17]. However, it was possible to obtain $T_m$ values for select 3-oligomer $\gamma$PNA systems at 5 µM concentration per oligomer. Melt curve experiments were repeated over two independent experiments for each condition.

**Nanostructure assembly**. Individual $\gamma$PNA, DNA, or aeg-PNA strands were added to different solvent conditions at 500 nM final concentration per oligomer based on the conditions for the study mentioned in the text and annealed using a Bio-Rad C1000 thermal cycler (Supplementary Table 3) by decreasing the temperature from 90 to 70 °C over 200 min, from 70 to 40 °C over 900 min, from 40 to 20 °C over 200 min, and finally holding at 4 °C.

**TIRF imaging**. Nanofibers were imaged at ×60 and ×90 magnification on a Nikon Ti2 microscope equipped with a ×60 1.4 NA Plan-Apo oil-immersion objective, ×1.5 magnifier, Prime 95B sCMOS camera (Photometrics), Nikon Perfect Focus System, and Nikon NIS-Elements software. To create flow chambers, channels ~3 mm apart were made with double-sided tapes on a glass slide. The coverslips were coated with 0.1% collodion in amyl acetate (EMS). Nanofibers were then immobilized to the coverslip surface at room temperature as follows. Biotinylated bovine serum albumin (BSA) at 0.1 mg mL$^{-1}$ in 1× PBS was incubated for 2–4 min. Excess biotin BSA was washed out, and the surface was incubated with BSA (10 mM dithiothreitol + 1 mg mL$^{-1}$ BSA in 1× PBS) for 2 min. Next, streptavidin at 0.1 mg mL$^{-1}$ in BSA solution was incubated for 2–4 min. Excess streptavidin was washed out with BSA. In the case of polar-aprotic solvent mixtures, the flow chamber was then washed with the same solvent mixture as the sample. Finally, 15 µL of nanofibers was added and incubated for 3–5 min. Excess nanofibers were washed out of the chamber with 1 mM Trolox.

**TEM imaging**. Four µL of annealed nanofibers sample was introduced on to a formvar-coated copper TEM grids. After 15 s, the solution was wiped off, and the sample was stained by addition of 1% uranyl acetate (EMS) and incubated on the sample for 5 s. Samples were imaged at 80 keV on a Joel JEM1011 TEM using the Advanced Microscopy Techniques capture engine software.

**Statistics and reproducibility**. Melting temperature experiments were repeated twice over two independent experiments for each solvent condition. Several images were acquired from different locations over repeated experiments of the microchannel and TEM grid surfaces to ensure reproducibility of the results as mentioned in each figure legends for TIRF and TEM experiments. Images for TIRF and TEM were analyzed using the Fiji (ImageJ), Microsoft Excel, and Minitab softwares. For TIRF images, we applied a threshold of 2 µm to exclude visible aggregates and optical distortions caused by DMSO to identify visibly separate nanofibers for length analyses. Width analyses performed on TEM images report median widths and interquartile ranges.

**Reporting summary**. Further information on research design is available in the Nature Research Reporting Summary linked to this article.

## Data availability

The source data that support the findings of Figs. 1c, 2b, d, and 4c; Supplementary Figs. 1, 2, 3a, 7, and 9; and Supplementary Table 3 are included in the Source data file. In addition, data repository that include raw and processed TIRF images for different conditions can be accessed from https://doi.org/10.6084/m9.figshare.12269288. Data repository that include raw TEM images can be accessed from https://doi.org/10.6084/m9.figshare.12269297. Datasets shared are labeled using the classification "YYYYMMDD–Experiment type–Experiment number–Sample condition." Any additional data are available from the corresponding author upon reasonable request.

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

## Acknowledgements

This work was supported in part by National Science Foundation grant 1739308 and by the Air Force Office of Science Research grant number FA9550-18-1-0199. We would also like to thank Dr. Catalina Achim, Dr. Bruce Armitage, Dr. Dilhara Jayarathna, and Ian Mitchell Harmatz for helpful discussions regarding oligomer functionalization and image analysis. γPNAs were a generous gift from Dr. Tumul Srivastava of Trucode Gene Repair, Inc. We would also like to thank Joseph Suhan, Mara Sullivan, and Center for Biological Imaging for their assistance in the collection of TEM data.

## Author contributions

R.E.T. designed the structural motif and computationally determined the oligomer sequences. R.E.T and S.K. together conceived and designed the experiments. A.P. and S.K. worked together on synthesis and characterization of aeg-PNA experiments. S.K. performed the experiments. R.E.T., S.K., and Y.L. performed the analysis and wrote the manuscript.

## Competing interests

The authors declare no competing interests.
