## [Peer Review File · Nature Communications]

Reviewers' comments:

Reviewer #1 (Remarks to the Author):

In this paper Kumar et al. reports a study on self-assembly of PNA higher order nano-structures in polar organic solvents. Overall it is an interesting study primarily based on TEM images. Although, fibers are detected and analyzed, the limited resolution only provides very indirect evidence for formation of the designed structures interpreted as "aggregated" monomer PNA fibers. Thus the results are quite preliminary. In addition the possible impact is "oversold" and some rationales and conclusions seem to lack support.

Specifically:

1. The authors state that DMF and DMSO prevent hybridization, which is not correct (ref 24 and the authors' own results) and refs 13-15 do not address this either. The authors may confuse DMF with formamide?
2. The major new progress is claimed to be self-assembly in organic solvents. However, the authors use very polar organic (DMF, DMSO, dioxane) solvents, which were already shown in 2007 (as also mentioned (ref24)) not to dramatically affect PNA-PNA hybridization, and in which PNA oligomers are also soluble. Thus the results are more or less as expected.
3. It is claimed that the gamma-PNA modification is necessary for the self-assembly due to "improved hydrophilicity". First of all without comparative data on pure aegPNA oligomers superiority of gammaPNA cannot be claimed. Secondly, it seems illogical to argue that "improved hydrophilicity" should be advantageous for moving from aqueous to organic solvents.
4. It is also claimed (p14) that the "conformation-enhancing modifications of γ PNA provide the high cooperativity, thermal stability, and specificity that are needed to enable PNA nanotechnology". Not correct without aegPNA data. These should be provided.
5. It is claimed that gammaPNA duplexes have higher stability than aegPNA duplexes, but references or evidence seems to be missing.
6. The T_m curves (fig. 1 C show multiple transitions. How was T_m determined? Also the y-axis should show absolute absorbance as the hyperchromicity is important to evaluate significance and structure. T_m curves of single stranded oligos should also be included in Suppl.
7. HPLC and MS data on the PNAs should be included in Suppl.
8. P4 gammaPNA is ascribed "exceptionally high affinity". The authors should consider LNA.
9. The discussion is over-speculative: "These γ PNA-based nanomaterials have transformative potential to extend the capability of nucleic acid nanotechnology, assisting in the transition of structural nucleic acid nanosystems out of the lab and into the clinic and into the manufacturing realm".
10. P 15: "nanostructures can be formed in solvent mixtures that promote hydrogen bonding" Which solvents "promote hydrogen bonding" and how?
11. P15: "The properties of γ PNA nanotechnology arise from the interaction of the chemical modifications of the strands and the structural motifs used to define the nanostructure". What does this mean? Please be scientifically specific.
12. P15: "In addition, the mechanics of these hybrid systems may for useful for triggering shape change and tuning thermal stability". What is the evidence/rationale for this postulate?

Other points

Title: The title promised programmable self-assembly, but only one structure is analyzed

P2: It is stated that high salt is necessary for DNA nano-structure formation. Is 20-100 mM high salt?

P5: the original ref (Nat. Struct. Biol, 4, 98 (1997) should be used instead of ref 28.

In terms of complex PNA structures, the authors may consider JACS 127, 1425 (2005).

Reviewer #2 (Remarks to the Author):

The authors, R. E. Taylor et al present a technical paper which was discussed the fabrication of 3-helix nanotubes comprised of either γ PNA oligomers or γ PNA-DNA hybrids. Even though the property and synthesis methodology of PNA molecules have been established by extensive studies, the construction of an artificial nanostructure made from PNA have been examined in only a few reports. Although the manuscript discussed interesting features of self-assembly of γ PNA and γ PNA-DNA hybrids, it would be good to address following issues accordingly.

1. In abstract, authors said, "Unlike the diameter-monodisperse populations of nanotubes formed using analogous DNA approaches, PNA structures appear to form bundles of nanotubes." Do authors express just phenomenological findings? Is it impossible to construct the diameter-monodisperse populations of nanotubes with PNA? If so, How?

2. Although authors wrote "programmable" (self-assembly of gamma-modified peptide nucleic acids in organic solvent mixtures) by means of various physical parameters such as solvent solution, strand substitution with DNA etc., I do not see significance of "programmability" of construction of various structures. I do just see some differences of structural morphologies by changing parameters. I think, programmable self-assembly discussed in this manuscript is very limited.

3. It is very important to verify final structures made of γ PNAs. Authors showed line profile scan in fig 2. How can we determine the final structures? Can you estimate final production yield? What about height? Why 3-helix nanotubes made of γ PNAs (which do not have net charges) stack each other along the duplex direction? Do you think stackings are related with an entropic depletion force? Do you have any possibilities to construct individual 3-helix nanotubes and visualize them with TEM?

4. Topology difference between PNA-DNA hybrid nanotubes with contiguous DNA and PNA-DNA hybrid nanotubes with crossover DNA are pretty much observed. What makes them difference? Is it related with persistence length of DNA and PNA?

5. Authors constructed one-dimensional (1D) PNA-DNA hybrids using the single stranded tile scheme. Construction of 2D by PNA-DNA hybrids will be somewhat different. Can you comment on it.

6. Depend upon solvent, such as 75% DMF: H₂O, 40% 1,4-dioxane: H₂O, and 75% DMSO:H₂O, PNA oligomers show different morphologies. If so, can you predict PNA structures with other solvent, or mixed? In order to be programmable self-assembly of γ PNA, it would be good to discuss feasibility to construct similar topological γ PNA structures with different organic solvent mixtures.

Reviewer #3 (Remarks to the Author):

This work proposes a method to realize Watson-Crick base pairing in harsh environments by using gamma-PNA alone or together with DNA instead of using DNA alone. To prove the concept, the authors adopted gamma-PNA from previous studies in the literature and confirmed that they form helical filaments and hybridize with DNA in several organic solvents. When it comes to forming hybrid nanotubes with DNA, it could be either similar to gamma-PNA alone or very different from that, leaving a new area to be explored. I think, however, whether the extraordinary programmability and subnanometer-precision structural predictability of DNA self-assembly could be fully harnessed with gamma-PNA (or its hybrids with DNA) still remains unanswered (which will require further studies in future). Overall, this work has presented solid experimental comparison on synthesis of nanotubes with gamma-PNA. I recommend this manuscript for publication after minor revisions.

Comments:

1. As discussed in the introduction, gamma-PNA may be a possible solution if one does not want to add protection to existing DNA origami towards applications of nanostructures in aprotic solvent environments. However, gamma-PNA structures appear to form bundles of nanotubes only; is it possible to build a structure similar to DNA origami with arbitrary geometry using gamma-PNA (for example, smiley face from gamma-PNA origami)?

2. For the control experiment replacing the entire system with the sequence-identical DNA oligomers, since the helicities (of gamma-PNA and DNA) are different, it does not seem to be a fair comparison. Is there any way to adjust the helicity of DNA to the value of gamma-PNA's so that the comparison between those two can be more revealing? On the flip side, is it possible to tune the helicity of PNA so that the shape of hybrid nanotubes can be controlled on purpose? The helicity seems to be very important as crossover DNA replacements created more differences.

Reviewers' comments:

Reviewer #1 (Remarks to the Author):

In this paper Kumar et al. reports a study on self-assembly of PNA higher order nano-structures in polar organic solvents. Overall it is an interesting study primarily based on TEM images. Although, fibers are detected and analyzed, the limited resolution only provides very indirect evidence for formation of the designed structures interpreted as “aggregated” monomer PNA fibers. Thus the results are quite preliminary. In addition the possible impact is “oversold” and some rationales and conclusions seem to lack support.

Specifically:

1. The authors state that DMF and DMSO prevent hybridization, which is not correct (ref 24 and the authors' own results) and refs 13-15 do not address this either. The authors may confuse DMF with formamide?

We thank our reviewer for this critical comment. We have revised this statement both in the abstract and the introduction section (lines 38-41) to “... dimethyl formamide (DMF) and dimethyl sulfoxide (DMSO) that can cause denaturation, reduced thermal stability and conformational changes in DNA duplexes.” And have cited references – 13, 17 and 18 in our manuscript.

2. The major new progress is claimed to be self-assembly in organic solvents. However, the authors use very polar organic (DMF, DMSO, dioxane) solvents, which were already shown in 2007 (as also mentioned (ref24)) not to dramatically affect PNA-PNA hybridization, and in which PNA oligomers are also soluble. Thus, the results are more or less as expected.

We thank our reviewer for this observation. As our reviewer mentions, Sen et al. (now, ref 18) has shown the solubility of Lysine-tagged PNA in DMF and dioxane. And has predicted the conditions for solubility in DMSO. They had also shown very little hinderance to duplex stability as part of this study. However, the novelty of our study comes from the idea that despite the stability of duplex formation, there are other considerations to allow for robust nanostructure formation. For instance, kinetic traps, misfolding and differences in solubility of complex nanostructures vs duplexes are a few things to be taken into consideration.

Our data highlights the idea that despite robust duplex formation and solubility of 2-oligomer systems in all the solvents (in varying v/v percentages with water) our reviewer has mentioned, complex, micron-scale nanostructures like the nanotube structures we have created can aggregate. Our aeg-PNA supplementary data (figure S3) and the section on γ PNA-DNA hybrids (figure 3) demonstrates that there might be a need for more thorough studies to allow simple unmodified aeg-PNA to form such structures.

Additionally, our specific focus on γ -modified PNA is aimed at highlighting both a design strategy to expand the single-stranded tile (SST) strategy to PNA space as well as the future potential of tunability of the side-chain modification to allow other interesting chemical functionalization. We believe our work and the recently published work from the Heemstra group (ref 66) taken in conjunction provide exciting new opportunities to be considered where one could form a bridge connecting the nucleic acid and peptide self-assembly fields.

3. It is claimed that the gamma-PNA modification is necessary for the self-assembly due to “improved hydrophilicity”. First of all, without comparative data on pure aegPNA oligomers superiority of gammaPNA cannot be claimed. Secondly, it seems illogical to argue that

“improved hydrophilicity” should be advantageous for moving from aqueous to organic solvents.

We thank our reviewer for this critical comment. We agree that we need to demonstrate performance of pure aeg-PNA versus gamma-PNA. To address this comment, we have performed sequential replacement of up to 7 aeg-PNA sequences in our gamma-PNA nanostructure, and in multiple cases replacement with more than 4 aeg-PNA oligomers prevents formation of distinct structures. We have provided both the self-assembly data on increasing aeg-PNA content in our supplementary information (figure S3b and c) and in our manuscript in lines 178-183, 229-239, 243-250 and 296-301.

We had previously focused on the “improved hydrophilicity” to motivate future work towards robust structure formation in aqueous environments but we agree with our reviewer’s comments and have edited out the words in the main text.

4. It is also claimed (p14) that the” conformation-enhancing modifications of γ PNA provide the high cooperativity, thermal stability, and specificity that are needed to enable PNA nanotechnology”. Not correct without aegPNA data. These should be provided.

We thank reviewer for this useful feedback. We have carried out the aeg-PNA experiments as mentioned above in lines 178-183, 229-239, 243-250 and 296-301 and have shown the figure in our supplementary information (figure S3). As mentioned in our discussion section (lines 296-301), that despite more than 50% aeg-PNA content showing aggregate formation, we believe the aeg-PNA still have the potential for nanostructure formation and would require a separate focus on the development of such a strategy. Hence, we have reworded our discussion section lines 294-296 to read “conformation-enhancing modifications of γ PNA provide the high cooperativity, thermal stability, and specificity that enable robust nanostructure formation.”

5. It is claimed that gammaPNA duplexes have higher stability than aegPNA duplexes, but references or evidence seems to be missing.

We thank our reviewer for their comment. We have carried out melt curve experiments of sequence-identical aeg-PNA and gamma-PNA to compare thermal stability (Figure S3a and Figure 1c). The corresponding lines in the manuscript are lines 136-146 and have also cited Sahu et al. (ref 47) to support this statement where they had based this observation on PNA-DNA and PNA-RNA duplex stability.

6. The T_m curves (fig. 1 C show multiple transitions. How was T_m determined? Also the y-axis should show absolute absorbance as the hyperchromicity is important to evaluate significance and structure. T_m curves of single stranded oligos should also be included in Suppl.

We would like to thank our reviewer for this critical comment. We agree with our reviewer’s insight in that melt curves for 3-oligomer systems show multiple transitions. We reran the melt curve experiments for these systems (both anneal and melt. We have shown the melt/heating curves in our manuscript.) and notice a repeated double-transition in the case of the aqueous buffer condition. This complex melting behaviour and the difficulty in obtaining a suitable lower baseline decreased our confidence in the thermodynamic analysis, so we removed this from the manuscript.

T_m was determined using the peaks of the first derivative (dA/dT). We have also now shown our Melt curves with the y-axis as normalized absorbance to help quantify hyperchromicity (figure 1c, Supplementary figures S1, S2, S3a). Supplementary table S3 now shows updated T_m values. T_m curves of the corresponding ss- γ PNA have also been included (Supplementary figure S2). The lines corresponding to this comment are lines 119-125.

7. HPLC and MS data on the PNAs should be included in Suppl.

We thank our reviewer for their useful inputs. We have included the HPLC and MS data for the aeg-PNA in our supplementary information (figure S11 and S12). We had purchased the gamma PNA strands from a commercially manufactured source in small scale batches. Hence, we have only shown the MS data for these oligomers in our supplementary information. (figure S13)

8. P4 gammaPNA is ascribed “exceptionally high affinity”. The authors should consider LNA. We thank our reviewer for this suggestion. We have considered gammaPNA for two main reasons – 1) because of its independence from counter-ion balance to form nanostructures and 2) the future capabilities to tune both the density and nature of the gamma-functionalization to either aid robust structure formation in water or other novel chemical functionalization such as amino-acids as ref 66 has shown. We believe for these reasons we did not want to consider LNA.

9. The discussion is over-speculative: “These γ PNA-based nanomaterials have transformative potential to extend the capability of nucleic acid nanotechnology, assisting in the transition of structural nucleic acid nanosystems out of the lab and into the clinic and into the manufacturing realm”.

We thank our reviewer for their critical feedback. We have revised these lines in our discussion section (lines 338-339) to now read “These γ PNA-based nanomaterials have transformative potential to extend the capability of nucleic acid nanotechnology towards more diverse nanosensing and biomedical applications.”

10. P 15: “nanostructures can be formed in solvent mixtures that promote hydrogen bonding” Which solvents “promote hydrogen bonding” and how?

We thank our reviewer for their question. We have edited this line because of its vagueness and it now reads “This work provides a proof of concept demonstration that micron-scale all- γ PNA and hybrid γ PNA-DNA nanostructures can be formed in organic solvent mixtures that meet the following criteria: (1) solvent mixtures are sufficiently polar to retain solubility of the PNA-DNA complexes even above 50% organic solvent; (2) solvent mixtures are aprotic or otherwise have reduced hydrogen-bonding donor/acceptor activity of the solvent to promote nucleobase hydrogen-bonding; and (3) solvent mixtures have a high boiling point to allow for slow thermal ramp annealing in the context of multiple unique oligomers to avoid kinetic traps or misfolding.” These are associated with lines 329-335 in our main text, citing references 18 and 58.

We write these lines in reference to our solvents here such as DMF and DMSO to aid any future work that focuses on solvent optimization and biophysical studies associated with such work. But additionally, solvents like acetonitrile, dichloromethane could be in consideration as well. The reason we did not mention solvents such as these is because acetonitrile, dichloromethane have boiling points lower than 100°C and they evaporate

during the annealing process, but this, however, doesn't eliminate the option of using them as part of a 3- and 4-solvent mixture to aid structure formation robustly.

11. P15: "The properties of γ PNA nanotechnology arise from the interaction of the chemical modifications of the strands and the structural motifs used to define the nanostructure". What does this mean? Please be scientifically specific.

We thank our reviewer for these editorial comments. We have revised these lines which now read "This study has focused on the design and formation of micron-scale filaments using a specific SST structural motif and specific type and density of mini-PEG γ -modifications on PNA. Therefore, future studies are needed to investigate additional structural motifs and the effects of type and density of gamma functionalization on these systems to expand towards a 3D architectural space. One example of this includes recent work from the Heemstra group on the self-assembly of micellar architecture using γ PNA with specific amino acid γ -modifications showing the capability to encode bilingual behavior through protein and nucleobase codes towards more complex nanoarchitectures." The associated line numbers are 310-317 with ref. 66.

12. P15: "In addition, the mechanics of these hybrid systems may be useful for triggering shape change and tuning thermal stability". What is the evidence/rationale for this postulate?

We thank our reviewer for this question and have edited these the text starting at line 319 to be more descriptive. "In addition, understanding the mechanics of these hybrid systems may be useful for implementing conformation-switching nanostructures through strategies such as toehold-mediated strand displacement already shown in DNA nanostructures. Future opportunities for responsive γ PNA nanosystems should take advantage of the uniquely tunable biorthogonality of left-handed γ PNA and potential for strand-displacement-driven switchability driven by high-affinity γ PNA." We have cited ref. 67 for dynamic DNA nanostructures, and ref. 68 for the helical handedness switching capability.

This general shape switching interest is motivated by current DNA nanotechnology mentioned in our introduction section as well as ref.67, where twisted and bent architectures allow for more complex 3D structures which aid nanosensing applications. Hence, understanding the mechanical parameters such as bending modulus, twist modulus and stretch modulus associated with γ PNA and hybrid duplexes in the context of nanostructure formation would be important to make diverse functional structures. Of particular interest is the finding that strand replacement with contiguous DNA sequences can lead to straight, apparently stiffened structures. In future work we will investigate this "stiffening" mechanism to determine if it is length-dependent and understand why apparent stiffening was observed only with contiguous strand replacement and not crossover replacement. If this process can be understood, stiffening the outside of a nanostructure might be a useful strategy to stabilizing both all-DNA and all-gammaPNA nanostructures.

Other points

Title: The title promised programmable self-assembly, but only one structure is analysed

We thank reviewer for their critical comment. We agree with their analysis and hence have changed our title to use the phrase "modular self-assembly". Programmability of structures would be a subsequent focus for our group as it would require redesigning the SST motif.

P2: It is stated that high salt is necessary for DNA nano-structure formation. Is 20-100 mM high salt?

We thank reviewer for their critical feedback. 20-100 mM of monovalent cations are typically in accordance with physiological conditions. However, typical DNA origami and SST assemblies use divalent Mg^{2+} buffers with concentration ranging between 12.5-25 mM which lie in the supraphysiological ranges. The corresponding reference in our manuscript would be ref. 10.

P5: the original ref (Nat. Struct. Biol, 4, 98 (1997) should be used instead of ref 28. In terms of complex PNA structures, the authors may consider JACS 127, 1425 (2005).

We would like to thank our reviewer for directing us to this reference. We completely agree and have incorporated this reference as ref.57 and occurs in lines 97-98 in our manuscript.

Reviewer #2 (Remarks to the Author):

The authors, R. E. Taylor et al present a technical paper which was discussed the fabrication of 3-helix nanotubes comprised of either γ PNA oligomers or γ PNA-DNA hybrids. Even though the property and synthesis methodology of PNA molecules have been established by extensive studies, the construction of an artificial nanostructure made from PNA have been examined in only a few reports. Although the manuscript discussed interesting features of self-assembly of γ PNA and γ PNA-DNA hybrids, it would be good to address following issues accordingly.

1. In abstract, authors said, “Unlike the diameter-monodisperse populations of nanotubes formed using analogous DNA approaches, PNA structures appear to form bundles of nanotubes.” Do authors express just phenomenological findings? Is it impossible to construct the diameter-monodisperse populations of nanotubes with PNA? If so, How?

We would like to thank our reviewer for this insightful question. We have shown it to be possible to reduce bundling effects of our PNA structures by using conventional anionic surfactant SDS in concentrations of 5.25 mM during self-assembly. The corresponding lines in our manuscript would be 258-291 along with figure 4 and figure S10.

2. Although authors wrote “programmable” (self-assembly of gamma-modified peptide nucleic acids in organic solvent mixtures) by means of various physical parameters such as solvent solution, strand substitution with DNA etc., I do not see significance of “programmability” of construction of various structures. I do just see some differences of structural morphologies by changing parameters. I think, programmable self-assembly discussed in this manuscript is very limited.

We thank our reviewer for their critical comment. We agree with their analysis and hence have changed our title to “Modular self-assembly”. Programmability of structures would be a subsequent focus for our group as it would require redesigning the SST motif to accommodate multiple different structures.

3. It is very important to verify final structures made of γ PNAs. Authors showed line profile scan in fig 2. How can we determine the final structures? Can you estimate final production yield? What about height? Why 3-helix nanotubes made of γ PNAs (which do not have net charges) stack each other along the duplex direction? Do you think stackings are related with an entropic depletion force? Do you have any possibilities to construct individual 3-helix nanotubes and visualize them with TEM?

To address this important question, we investigate the application of the anionic surfactant SDS to reduce aggregation starting on line 258. As shown in Figure 4, width polydispersity is reduced when SDS is used. While we did not obtain purely monodisperse populations of nanotubes, we substantially reduced the median width of gammaPNA nanostructure, and in TEMs of these structures, numerous filaments in the 10 nm and sub-10 nm width range are visible.

From our SDS studies, it is not possible to understand if bundling is feature of growth or if entropic depletion forces drive the bundling as a lower energy state.

4. Topology difference between PNA-DNA hybrid nanotubes with contiguous DNA and PNA-DNA hybrid nanotubes with crossover DNA are pretty much observed. What makes them difference? Is it related with persistence length of DNA and PNA?

We would like to thank our reviewer for this question. We have addressed this in the manuscript in lines 226-250. While we do not clearly understand this phenomenon at this point, one possible interpretation comes from findings from ref. 47 that in the case of contiguous DNA replacements, DNA oligomers undergo conformational change as they interact with the preorganized complimentary γ PNA. This torsion of the DNA may lead the gammaPNA-DNA segments to take on a P-form type structure. However, in our case, this behaviour appears to be dependent on segment length or other hydrophobic effects, as demonstrated by the aggregation and less straight appearing filaments formed with DNA crossover replacements. Indeed, hydrophobic effects do seem to play a role given the increased aggregation of both aeg-PNA and DNA hybrids and subsequent decreased aggregation in the presence of SDS with respect to all- γ PNA nanostructures (lines 273-280).

5. Authors constructed one-dimensional (1D) PNA-DNA hybrids using the single stranded tile scheme. Construction of 2D by PNA-DNA hybrids will be somewhat different. Can you comment on it.

The topology of 2-domain SST systems is necessarily simple where each 2-domain SST is capable of binding and connecting exactly two other domains. This enables a linear-type building motif and does not accommodate branching. To create 2D nanostructures, 3- or 4-domain SSTs will be needed. For gammaPNA, this will require longer oligomers (e.g. $3 \times 5 = 15$, $3 \times 6 = 18$, $4 \times 5 = 20$ bases). Alternately, if long scaffold DNA is used, short 2-domain gammaPNAs may suffice as staples. However, to create 2D structures, it will be necessary to create designs that use crossover spacing equal to the hybrid number of bases per turn (i.e. 16 DNA-gammaPNA hybrid base pairs per turn rather than 10.5 DNA bases or 18 PNA bases). Therefore gammaPNA staples that are 8 and 16 bases in length might be useful for creating origami-type hybrid systems. This is an approach that we can investigate in future studies, but future studies will also be needed to better understand the helical properties of hybrid structures as a function of domain length and the impacts of other structural motif characteristics like crossover features on hybrid system morphology and mechanics.

6. Depend upon solvent, such as 75% DMF: H₂O, 40% 1,4-dioxane: H₂O, and 75% DMSO:H₂O, PNA oligomers show different morphologies. If so, can you predict PNA structures with other solvent, or mixed? In order to be programmable self-assembly of γ PNA, it would be good to discuss feasibility to construct similar topological γ PNA structures with different organic solvent mixtures.

We thank our reviewer for their question. We have edited the corresponding line to be more descriptive and it now reads "This work provides a proof of concept demonstration that micron-scale all- γ PNA and hybrid γ PNA-DNA nanostructures can be formed in organic solvent mixtures that meet the following criteria: (1) solvent mixtures are sufficiently polar to retain solubility of the PNA-DNA complexes even above 50% organic solvent; (2) solvent mixtures are aprotic or otherwise have reduced hydrogen-bonding donor/acceptor activity of the solvent to promote nucleobase hydrogen-bonding; and (3) solvent mixtures have a high boiling point to allow for slow thermal ramp annealing in the context of multiple unique oligomers to avoid kinetic traps or misfolding." These are associated with lines 329-335 in our main text, citing references 18 and 58.

We write these lines in reference to our solvents here such as DMF and DMSO to aid any future work that focuses on solvent optimization and biophysical studies associated with

such work. But additionally, solvents like acetonitrile, dichloromethane could be in consideration as well. The reason we did not mention solvents such as these is because acetonitrile, dichloromethane have boiling points lower than 100°C and they evaporate during the annealing process, but this, however, doesn't eliminate the option of using them as part of a 3- and 4-solvent mixture to aid structure formation robustly.

Reviewer #3 (Remarks to the Author):

This work proposes a method to realize Watson-Crick base pairing in harsh environments by using gamma-PNA alone or together with DNA instead of using DNA alone. To prove the concept, the authors adopted gamma-PNA from previous studies in the literature and confirmed that they form helical filaments and hybridize with DNA in several organic solvents. When it comes to forming hybrid nanotubes with DNA, it could be either similar to gamma-PNA alone or very different from that, leaving a new area to be explored. I think, however, whether the extraordinary programmability and subnanometer-precision structural predictability of DNA self-assembly could be fully harnessed with gamma-PNA (or its hybrids with DNA) still remains unanswered (which will require further studies in future). Overall, this work has presented solid experimental comparison on synthesis of nanotubes with gamma-PNA. I recommend this manuscript for publication after minor revisions.

Comments:

1. As discussed in the introduction, gamma-PNA may be a possible solution if one does not want to add protection to existing DNA origami towards applications of nanostructures in aprotic solvent environments. However, gamma-PNA structures appear to form bundles of nanotubes only; is it possible to build a structure similar to DNA origami with arbitrary geometry using gamma-PNA (for example, smiley face from gamma-PNA origami)?

We would like to thank our reviewer for these important questions. We have addressed the bundling issues of our constructs using conventional surfactants such as SDS in concentrations of 5.25 mM during self-assembly. The corresponding lines in our manuscript would be 258-291 along with Figure 4 and Figure S10.

We currently envision polyhedral structures (tetrahedra, octahedra) as well as ring structures for gamma-PNA as subsequent focus within our group. The smiley-face structure made using DNA origami used a scaffolded approach using plasmid M13 DNA, which would not be possible with PNA, because PNA synthesis is limited to making up to 22-25 bases with acceptable yields. However, we do believe that a 3- or 4-domain SST system could be designed using gammaPNA that would be able to assemble into arbitrary 2D and 3D nanostructures like a smiley-face using an approach like DNA Lego Bricks. However, this approach would require a vast number of distinct gammaPNA oligomers, which at this time might be prohibitively expensive.

2. For the control experiment replacing the entire system with the sequence-identical DNA oligomers, since the helicities (of gamma-PNA and DNA) are different, it does not seem to be a fair comparison. Is there any way to adjust the helicity of DNA to the value of gamma-PNA's so that the comparison between those two can be more revealing? On the flip side, is it possible to tune the helicity of PNA so that the shape of hybrid nanotubes can be controlled on purpose? The helicity seems to be very important as crossover DNA replacements created more differences.

We would like to once again thank for the reader for this insightful comment, because research into the control of helicity and twist in these structures is part of our future research plan. It would be challenging to perform a more "apples-to-apples" comparison of SST nanotubes made of DNA versus gammaPNA. The challenge arises due to the combination of PNA's higher binding affinity and reduced rotation-per-base. To answer their question, 3-helix DNA nanotubes can be formed using traditional SST approaches, but would require domains longer than 6 bases for stability while allowing for the structurally similar net 240 degrees of rotation used to form the 3-helix gammaPNA. For example, 18 base long DNA

oligomers would each to allow for approximately a net 240-degree rotation. However, to enable crossovers that do not result in net twist, each repeating unit would need to be a multiple of 10.5 bases long. Therefore one such system might need 3 helices x 2 hybridizing strands x 21 bases = 126 bases, which does not divide evenly by 18. A DNA system would therefore have some 18-base strands and some 24 base-strands, and therefore even this improved design would not provide a fair comparison.

Our initial expectation was that indeed, the introduction of DNA would twist the structure if introduction of DNA into the system introduced regions of DNA-gammaPNA duplexes with a more tightly twisted helix with 16 bases per turn rather than 18 bases per turn as reported in the literature. Given that we do not observe twist in DNA hybrid studies, we will be looking at simpler, non-repeating structures in the future to study this behaviour (including mechanics). For now, we observe, surprisingly that introduction of DNA may actually stiffen the structure, potentially due to reorganization of DNA due to high torque application. Our working hypothesis is that DNA-gammaPNA behaviour is a function of solvent solution, gamma-modification positions, and structural motif (including strand purpose crossover/contiguous). At this point we hypothesize that crossover strands have 6 bases on each helix, and perhaps reorganization of the helix as seen with contiguous replacement may be a DNA-gammaPNA interface length-dependent phenomenon only seen with longer interfaces like the contiguous DNA-gammaPNA segments.

Reviewers' comments:

Reviewer #1 (Remarks to the Author):

In the revised version the authors have addressed (almost all of) the specific points raised. However, the more general criticism is not addressed.

The study is still almost exclusively descriptive and phenomenological and no molecular structural information is provided for the self assembly fibers. Indeed the primary self assembled complexes are termed "tubes", without convincing evidence for a hollow structure. Also the diameter of the observed higher order fibers are ca. 20 nm (with a broad variation), suggesting that two, three and more of the assumed self-assembled helix structures proposed in Fig 1d are forming this fiber. However, the driving force(s) and structural basis for this is not experimentally addressed, and it is not clear how these results readily extends to other and more complex structures.

The introduction and in particular the (still too long) discussion is predominantly speculative beyond the experimental evidence, and really "oversell" the very narrow findings, in terms of future perspectives. For instance any comparison to origami.

Specific points:

1. HPLC analyses for the gamma PNA oligomers are missing in supplementary and the MS spectra of these show multiple peaks that are not explained indicating low purity.
2. It is discussed on page 15 that in the mixed PNA-DNA assemblies the DNA oligomers have to undergo conformational changes do to the large helical pitch of the PNA helix. The authors may have overlooked that the pitch of a (gamma)PNA/DNA helix is 15 bp/turn.
3. The comparison of the SDS effect of protein structure (p18) does not seem directly relevant, since SDS completely unfolds (denatures) the (complex) three dimensional protein structure by disrupting critical hydrophobic interactions.

Reviewer #2 (Remarks to the Author):

The authors answered the comments very nicely. The work is solid and informative, and the manuscript is well organized. I believe the work presented here is important and deserves publication.

Sung Ha Park

Reviewer #3 (Remarks to the Author):

My technical concerns were adequately addressed. No more comments.

Reviewer #1 (Remarks to the Author):

In the revised version the authors have addressed (almost all of) the specific points raised. However, the more general criticism is not addressed.

The study is still almost exclusively descriptive and phenomenological and no molecular structural information is provided for the self assembly fibers. Indeed the primary self assembled complexes are termed “tubes”, without convincing evidence for a hollow structure. Also the diameter of the observed higher order fibers are ca. 20 nm (with a broad variation), suggesting that two, three and more of the assumed self-assembled helix structures proposed in Fig 1d are forming this fiber. However, the driving force(s) and structural basis for this is not experimentally addressed, and it is not clear how these results readily extends to other and more complex structures.

We thank the reviewer for their useful feedback. We agree that an experiment such as cryoTEM could definitively verify the molecular structure/conformation of gamma-PNA double helices through Watson-Crick base-pairing in the nanostructure. This would however be a future focus within our group.

Our TEM studies currently show little evidence of stain occupying the inner circumference of our filamentous structures because of the nature of peptide-backbone staining using Uranyl acetate (with potential interactions of the sulfate anion in SDS to interact with the Uranium dioxide cation in Uranyl acetate). Therefore, the system might require high-resolution (HR-TEM) for further evidence of hollow structures, if it can be detected.

However, there is experimental evidence within our studies that verify the Watson-Crick driving forces and thereby the structural basis that verify our SST design and nanotube characterization. For instance, control experiments in the presence of a mismatched strand or in the absence of one contributing gamma-PNA oligomer with or without SDS, does not form extended nanostructures (lines 173-175, marked-up manuscript). Additionally, the experimental evidence of formation of gamma-PNA-DNA hybrid nanostructures indicate that the primary driving forces are still Watson-Crick since the backbone has now moved away from an aeg-peptide backbone to a sugar-phosphate backbone in up to 3 oligomers in the SST motif. So, any hydrophilic-hydrophobic based driving force for structure formation would have been affected considerably. We have thus added lines 258-260 to reflect the different driving forces. Our melting experiments also primarily show sigmoidal curves characteristic of Watson-Crick base-pairing. This driving force would therefore extend to more complex structures as well.

Furthermore, our SDS studies indicate that the “bundling” effects of gamma-PNA nanotubes is primarily due to non-specific hydrophobic interactions between 2, 3 or more self-assembled helical tubes. SDS in our case as we mention in specific comment 3 here, as well as in lines 268-280 in our manuscript disrupts non-specific hydrophobic interactions through the introduction of electrostatic interactions. The broad variation in diameters with our construct we believe is an effect of the micron-scale size of the formed constructs. Since we are in the mesoscale, post-structure formation, we believe any future complex structures that have micron-scale structure growth, would be affected by such non-specific interactions but could be rectified with an anionic surfactant such as SDS.

The introduction and in particular the (still too long) discussion is predominantly speculative beyond the experimental evidence, and really “oversell” the very narrow findings, in terms of future perspectives. For instance any comparison to origami.

We thank the reviewer for their editorial comments. We have addressed this concern by excluding and editing lines from our manuscript as indicated in lines 31-35, 42-49, 316-318, 320-322, 331-342, 353-358 (marked-up document) in our introduction section as well as our discussion section. Specifically, we have edited lines to reflect observations from our findings without comparisons to the field of DNA origami.

Specific points:

1. HPLC analyses for the gamma PNA oligomers are missing in supplementary and the MS spectra of these show multiple peaks that are not explained indicating low purity.

We thank the reviewer for their observation. We have recently received permission from our gPNA manufacturer to publish the LC-MS characterization of the gPNA sequences studied here. We have now included both chromatograms and ESI-spectra for all our gPNA sequences used.

We recognized one of the chromatograms representing our biotin-modified gPNA strand to have an additional shoulder in addition to the main peak [Supplementary figure S13-(f)]. We have attributed this to possible aggregation of our biotin-modified strand (S13 figure caption) since the corresponding ESI-MS spectra shows a single species matching the expected mass [Supplementary figure S14-(f)].

2. It is discussed on page 15 that in the mixed PNA-DNA assemblies the DNA oligomers have to undergo conformational changes do to the large helical pitch of the PNA helix. The authors may have overlooked that the pitch of a (gamma)PNA/DNA helix is 15 bp/turn.

We would like to thank the reviewer for this comment. As the reviewer mentions, Ref. 61 in our manuscript describes the pitch associated with gPNA-DNA to be 15 bp/turn in duplexes. We have added lines 237-243 to clarify this.

3. The comparison of the SDS effect of protein structure (p18) does not seem directly relevant, since SDS completely unfolds (denatures) the (complex) three-dimensional protein structure by disrupting critical hydrophobic interactions.

We thank the reviewer for their critical comment. As the reviewer clearly points out surfactants are known to cause denaturation in proteins, whereby secondary or higher-order structure is partially or completely disrupted. Therefore, it is logical that surfactants can also significantly alter the solution behavior of peptides, given that they are composed of the same elementary building blocks as their larger counterparts.

We were motivated however by the studies that have shown that unlike in large proteins, charged shorter peptides adopt secondary structure conformations (α -helix and β -sheet conformations) from unstructured conformations in the presence of oppositely charged surfactants like SDS through both the hydrophobic tail and electrostatic interactions. We had therefore hypothesized that SDS might not hinder Watson-Crick base pairing between nucleobases and have added lines 276-280. We have also added Ref 65 in addition to our previous Refs. 62-64 to highlight that SDS in the case of peptides does not hinder or in some cases, promotes secondary structure formation based on its tail molecule, charge and concentrations. Motivated by this phenomenon of SDS, we studied morphology regulation

our gPNA bundled structures. Particularly, we hypothesized that the use of SDS would not disrupt the hydrogen bonding in Watson-Crick base pairing of γ PNA oligomers and could promote reduced bundling over a specified range of concentrations

Reviewer #2 (Remarks to the Author):

The authors answered the comments very nicely. The work is solid and informative, and the manuscript is well organized. I believe the work presented here is important and deserves publication.

Sung Ha Park

Reviewer #3 (Remarks to the Author):

My technical concerns were adequately addressed. No more comments.

REVIEWERS' COMMENTS:

Reviewer #1 (Remarks to the Author):

My concerns have now been addressed, apart from the term "nanotube". Accuracy and concise terminology is a cornerstone in science. Therefore since no convincing evidence is available for the presence of a cavity in the "nanotube", the correct term should be the neutral "nanofiber".

Reviewer #1 (Remarks to the Author):

My concerns have now been addressed, apart from the term "nanotube". Accuracy and concise terminology is a cornerstone in science. Therefore since no convincing evidence is available for the presence of a cavity in the "nanotube", the correct term should be the neutral "nanofiber".

We thank our reviewer for their final feedback. We have now replaced the word "nanotube" with the neutral term 'nanofiber' within the manuscript.